# Research on Residual Stress of a BS700 Butt-Welded Box Section and Its Influence on the Stability of Axial Compression Members

**DOI:** 10.3390/ma13153282

**Published:** 2020-07-23

**Authors:** Xingkun Xie, Fei Shao, Lei Gao, Lixiang He, Linyue Bai

**Affiliations:** College of Field Engineering, Army Engineering University of PLA, Nanjing 210007, China; jfxie2020@sina.com (X.X.); he.l2020@aliyun.com (L.H.); baily016@sina.cn (L.B.)

**Keywords:** BS700 high-strength steel, welding box section, residual stress, axial compression, stability factor

## Abstract

BS700 high-strength steel is widely used in engineering. Welding residual stress during the manufacturing process has a significant influence on the structural safety and service life of steel structures. In this study, the residual stress of a BS700 butt-welded box section axial compression member was studied by the blind-hole method, its distribution law was summarized, and a residual stress distribution model was established. By establishing a finite element model considering initial geometric imperfection and residual stress, the influence of residual stress on the stability of axial compression members was analyzed. The results illustrated that the residual tensile stress near the weld in the welded box section axial compression members was the largest: the average residual tensile stress reached 76.6% of the measured steel yield strength, the residual tensile stress at the roof and web were almost the same, and the residual tensile stress at the corner was approximately 11.6% of the measured yield strength. The residual stress had a different influence on the stability factor of the axial compression members with different width-thickness ratios, and the influence decreased with the increase in the width-thickness ratio. In addition, when the slenderness ratio of members ranged between 20 and 70, the residual stress had a significant influence on the stability of members, while outside that interval, the influence was relatively small.

## 1. Introduction

Steel structures have the advantages of high strength and light weight, and are widely used all over the world in buildings, bridges, and roof-supporting systems. Compared with ordinary structural steel, high-strength steel has obvious advantages in terms of strength, ductility, weldability, and structural mechanical performance. The theoretical and experimental research on the mechanical properties and stability of welded structures made of high-strength steel has become a hot topic [1,2,3].

A large body of research has shown that residual stress has different effects on the mechanical properties of welded structures of high-strength steel [4,5,6]. A series of experiments and finite element numerical studies have shown that the residual stress is closely related to the section geometry, and a relevant residual stress distribution model was established [7,8,9,10]. Lee et al. [11] presented a sequentially coupled three-dimensional thermal, metallurgical, and mechanical finite element model to simulate welding residual stresses in a high-strength carbon steel butt weld considering solid-state phase transformation effects. Anas et al. [12] carried out experiments on the welding residual stress of P91 pipe after post-weld heat treatment and the finite element method of modelling residual stresses is presented. Lee et al. [13,14] conducted experiment and numerical simulation on welded high-strength steel thin-walled plate-to-plate joints and discussed the influence of some key welding parameters on the magnitude and distribution of residual stress. Trayana et al. [15] used data from the experiment and numerical results and the literature on welding residual stress distribution of steel members in which I-beams were analyzed. Systematic information about residual stress distribution and magnitude was obtained, and some guiding principles for statistical characterization of residual stress in welded components were proposed. Tuan et al. [16] studied a unified residual stress model applicable for welded high-strength steel’s I-beams. Experimental results confirmed that the compressive residual stress is associated with the geometric shapes of the cross-section and has nothing to do with the grade of the steel. Through experiments, theory, and numerical analysis, Kovesdi et al. [17,18,19] investigated the flexural buckling performance of a high-strength steel welded box section. The Ayrton-Perry formula-derived analytically—was modified; with the use of an adjusted buckling curve, the welding flexural buckling performance of the box column could be accurately predicted. Fang et al. [20] established a finite element model of a cold-formed high-strength steel pipe column and used the verified model to conduct a parametric study to determine the strength of the cold-formed steel pipe column with different section sizes, component slenderness ratios, geometric defects, and steel grades. Shi et al. [21,22,23,24,25,26] carried out a series of theoretical and experimental studies on the distribution of welding residual stress in high-strength steel and the mechanical performance of axial compression members, and obtained the distribution law of welding I-shaped and box-shaped section residual stress and its influence on the stability of compression members. Based on the test results, a finite element numerical model was established to analyze the parameters of the box-type axial compression member, and some suggestions on the design of a high-strength steel welded axial compression member were given by comparing European EN1993-1-1 [27] and China GB50017-2003 [28]. Li et al. [29,30] discussed the influence of welding residual stress by comparing the results obtained without accounting for the welding residual stress on the ultimate strength of stiffened plates under uniaxial compressive load by nonlinear finite element analysis. Xiao et al. [31] studied the microstructure and mechanical properties of welded joints of BS700MC high-strength steel used for automotive girders through crack sensitivity, micro-hardness, bending, impact, tensile test, and micro-structural observation.

Scholars around the world have carried out extensive research work on the welding residual stress distribution rule and its influence on the stability of the axial compression artifacts of all kinds of high-strength steel members of specific strength grades, but research on welding residual stress distribution and its influence on the stability of the members of the BS700 box section is very rare. Taking BS700 welded box section axial compression members as the research object, through experiments and numerical simulation, the welding member residual stress distribution rule and its influence on the stability of the axial compression member were analyzed.

This paper is organized as follows: details on the material properties and the residual stress test are presented in Section 2, along with a residual stress distribution model of a butt-welding box section. In Section 3, through a numerical simulation, the residual stress distribution law of welding members and its effect on the stability of the axial compression members are analyzed. In Section 4, some conclusions and suggestions for engineering are given.

## 2. Material Properties and Residual Stress Testing

In order to study the distribution of welding residual stress and its influence on the stability of the members, we carried out a high-strength steel material test and a residual stress test on the BS700 butt-welding box section specimen.

### 2.1. Material Test

The calibration specimens and test specimens were taken from a certain steel company batch production with the same batch of materials; the chemical composition is shown in Table 1 [31]. After mechanical processing, the specimens were annealed to eliminate stress.

The yield strength and tensile (ultimate) strength of BS700 high-strength steel were obtained through a uniaxial tensile test. The results of the test provided the basic mechanical properties of the materials for the numerical simulation. The preparation and tensile test of specimens followed the requirements of GB/T2975-1998 [32] and GB/T228.1-2010 [33]. Three specimens were designed and made in the experiment by using the fusion welding method. The total length, width, and thickness of the specimens were 200, 20, and 3 mm, respectively. Due to the influence of production errors, the specific size of each specimen was as shown in Table 2. The failure mode of the specimen is shown in Figure 1, and the stress-strain curve of the TS1 specimen is shown in Figure 2. The results of the material test are listed in Table 3. According to the test results, the specimens exhibited good ductility deformation and a significant necking phenomenon. The measured elastic modulus, yield stress, and ultimate tensile stress of the specimens were 207.4 GPa, 739.2 MPa, and 781.9 MPa, respectively. It should be pointed out that, due to the influence of an installation error, the failure of specimen TS2 presented as an oblique fracture, and the yield stress and ultimate tensile stress were relatively small compared with the other two specimens, TS1 and TS3.

### 2.2. Residual Stress Test

#### 2.2.1. Test principle

Residual stress is a kind of stress that is self-balanced in the member. The basic principle of the blind-hole method to test residual stress is to drill holes on the surface of the member to destroy the original equilibrium state in order to release stress. Stress release produces strain, and the residual stress can be calculated according to the measured strain [34]; the principle is shown in Figure 3.

In an isotropic material in a general state, there exists a residual stress field area. The maximum and minimum principal stress are *σ_1_* and *σ_1_*, respectively, on the surface of this area. We pasted a special strain rosette and made a hole in the center of it to allow stress release, which produces strain filaments. The residual stress can be calculated as follows:(1)$$\sigma_{1}=\frac{E}{4A}\left(\varepsilon_{1}+\varepsilon_{3}\right)-\frac{E}{4B}\sqrt{{\left(\varepsilon_{1}-\varepsilon_{3}\right)}^{2}+{\left(2\varepsilon_{2}-\varepsilon_{1}-\varepsilon_{3}\right)}^{2}}$$
(2)$$\sigma_{2}=\frac{E}{4A}\left(\varepsilon_{1}+\varepsilon_{3}\right)+\frac{E}{4B}\sqrt{{\left(\varepsilon_{1}-\varepsilon_{3}\right)}^{2}+{\left(2\varepsilon_{2}-\varepsilon_{1}-\varepsilon_{3}\right)}^{2}}$$
(3)$$\mathrm{tg}2\theta =\frac{2\varepsilon_{2}-\varepsilon_{1}-\varepsilon_{3}}{\varepsilon_{3}-\varepsilon_{1}}$$
where *ε_1_*, *ε_2_*, and *ε_3_* are the release strains in the three directions with a unit of *μ**ε*, σ_1_ and σ_2_ are the maximum and minimum principal stresses, respectively, in MPa; *θ* is the angle between σ_1_ and *ε_1_*, in degrees; *E* is the elastic modulus of materials with the unit of GPa; *A* and *B* are release coefficients.

The measurement and calculation were conducted according to the GB/T31310-2014 method (see Appendix B of Ref. [35]), using the low-speed drill blind-hole method to measure the near-surface residual stress of the material.

#### 2.2.2. Coefficient Calibration of Residual Stress A and B

A and B are release coefficients, which are generally determined by experimental calibration, with reference to the test method of GB/T31310-2014 (see Appendix A of Ref. [35]). The calibration test and determination of residual strength used the same batch strain rosette, and the perpendicular direction of the resistance strain gauge is consistent with the length and width direction of the calibration samples. The length of the calibrated specimen is 360 mm, the width at both ends is 90 mm, and the width at the middle is 60 mm. Two types of specimens, with a thickness of 3 or 6 mm, were made. The strain rosette on the calibration specimen was arranged as shown in Figure 4. The calibration specimen was first stretched without drilling, and then, stretched after drilling. The test process is shown in Figure 5.

Constants A and B should be measured twice at least. If the compressive error is bigger than 10%, the constants will be calibrated again. The constants A and B are obtained from two calibration processes, from which we took the average value, and the calculation equation is as follows:(4)$$A=\frac{\varepsilon_{1}+\varepsilon_{3}}{2\sigma }$$
(5)$$B=\frac{\varepsilon_{1}-\varepsilon_{3}}{2\sigma }$$

Substituting the measured strain into the above equation, the release coefficients *A* and *B* can be determined. The results are *A* = −0.1223 and *B* = −0.2045.

#### 2.2.3. Residual Stress Test

In order to obtain the distribution rule of residual stress in a section, the residual stress of four specimens was tested. Figure 6 shows the drawings of samples. The number and section size are listed in Table 4. The length of the specimen is not less than three times the width of the section; 1.5 times the width of the specimen is reserved at two ends of the measuring points to reduce the end effect and the adverse effect of uneven heat input in the welding process.

There are eight measuring points in the welded box section specimen RS1. Point 1 is on the weld line and point 2 is the heat-affected zone of the weld line corresponding to point 1, which is 20 mm away from the weld line. Point 8 is on the weld (53 mm away from the bend) and point 7 is the heat-affected zone of the weld corresponding to point 8, which is 20 mm away from the weld (33 mm away from the bend line). Points 3, 4, and 6 are the corners, and the distance from the edge is 5 mm. Point 5 is the middle position of the non-weld panel (33 mm away from the elbow). The relative positions of the measuring points of other specimens are the same as those for specimen RS1. The specific positions of the measuring points are shown in Figure 7. Residual stress measurement tests (JH-ZK Punch and JH30-Residual stress measure, Shandong, China) are shown in Figure 8, and the tested residual stresses at each measuring point are listed in Table 5.

*ε**_1_*, *ε**_2_* and *ε**_3_* are the strain corresponding to drilling at 90°, 45°, and 0°; *ε**_3_* is in the x direction, namely along the length direction of the x axis. *σ*_max_ is the maximum principal stress, *σ*_min_ is the minimum principal stress, and *β* is the angle between the maximum principal stress and the x axis. Meanwhile, the stress along the axial direction of the member, i.e., the stress along the x axis direction, is calculated by the following equation:(6)$$\sigma =\sigma_{\max}\cos(\beta )-\sigma_{\min}\sin(\beta )$$

### 2.3. Test Results and Analysis

Table 5 shows the residual tensile stress near the weld of the highest peak; the residual tensile stress peak values of the flange and web are almost equal, and smaller than the yield strength of the material. The average residual tensile stress peak value is 570.1 MPa, about 76.6% of the measured yield strength steel, and the maximum peak stress is 619.8 MPa. The residual tensile stress at the corner is 86.5 MPa, about 11.6% of the yield strength. The mean residual compressive stress is related to the width-thickness ratio. The larger the width-thickness ratio, the smaller the mean residual compressive stress.

Combining the experimental data with references [6,14,36], the residual stress distribution model shown in Figure 9 was proposed. Because the left and right sides of the cross-section are symmetrical, the residual stress distribution at the left and right sides is also assumed to be symmetrical, but only the residual stress distribution of the right side is shown. In Figure 9, $\sigma_{ft}$ represents the residual tensile stress at the weld position of the flange, $\sigma_{fc}$ represents the residual compressive stress at the flange, $\sigma_{fwt}$ represents the residual tensile stress at the cold corner, and $\sigma_{wc}$ represents the residual compressive stress of the web. a_1_, b_1_, c, a_2_, b_2_,c, a_2_, b_2_, d_1_, e_1_, g, e_2_, and d_2_ represent the length of each distribution area of residual stress. $\sigma_{ft}=\left(0.7∼0.85\right)f_{y}$, $\sigma_{fwt}=\left(0.05∼0.15\right)f_{y}$, $\sigma_{fc}=-\left(0.1∼0.3\right)f_{y}$, $\;\sigma_{wc}=-\left(0.1∼0.2\right)f_{y}$, $a_{1}=t∼2t$, $b_{1}=0.05b$ and $d_{1}=t∼2t$. The rest of the dimensions can be obtained according to the stress equilibrium conditions.

The residual tensile stress increases with the increase in the width‒thickness ratio, while the residual compressive stress decreases with the increase in the width‒thickness ratio. The residual stress and distribution length are given by the following equations:(7)$$\iint \sigma_{r}dA=0$$
(8)$$a_{1}+b_{1}+c+b_{2}+a_{2}=0.5b$$
(9)$$d_{1}+e_{1}+g+e_{2}+d_{2}=0.5h$$
where *σ_r_* is the residual stress, b is the width of the section, h is the height of the section, and *t* is the thickness of the section.

In order to verify the proposed residual stress model, RS1 and RS4 were selected for comparison. The relevant data are listed in Table 6. Comparing the test data with the residual stress distribution patterns, the results are shown in Figure 10. The dots are the test measurement values and the solid line is the residual stress model.

## 3. Finite Element Simulation and Result Analysis

### 3.1. The Establishment of a Finite Element Model

In this study, ANSYS software (Ansys 15.0) was used to establish a finite element analysis model to carry out numerical simulation. First, the basic finite element model was established considering the influence of residual stress and geometric defects. In order to consider the local buckling of plates and the overall buckling of rods, shell181 was used in the analysis model instead of the beam element. The bilinear isotropic strengthening model was applied to simulate the nonlinear behavior of the BS700 high-strength steel. The bilinear isotropic reinforcement constitutive model was adopted to simulate the BS700 high-strength steel. The elastic modulus was 2.07 × 10^5^ MPa; the yield stain and stress were 0.003398 and 700 MPa, respectively; the strain and stress at the end of yield stage were 0.0092 and 704.4 MPa, respectively; the ultimate tensile strain and stress were 0.0218 and 764.1 MPa, respectively. The hardening parameter was assumed to be 0.005, here. In order to simulate the rigid surrounding and impose constraints and load on the model, two rigid plates with a large elastic modulus were added to both ends of the member. When applying constraints, the node at the center of both plates was constrained. The basic finite element model that was established is shown in Figure 11. The finite element analysis of the members was carried out with and without residual stress, respectively.

The initial geometric defects were introduced into the model by the uniform defect mode method, and the eigenvalue buckling analysis was first adopted to obtain the local buckling modes of the specimens (as shown in Figure 12). Then, a certain coefficient was multiplied to the maximum displacement so that the maximum defect value was 1/1000 of the plate width. The nodes and elements were reestablished according to the sinusoidal half-wave curve, so that the maximum initial deformation in the middle of the member was 1/1000 the length of the member [37].

Nonlinear buckling analysis was carried out by gradually increasing the nonlinear static load to obtain the critical load when the structure becomes unstable, which is a kind of static analysis considering the large deformation effect. This analysis was carried out up to the ultimate load or maximum load of the structure. To determine the critical load of the structure, it was necessary to find the critical value $p_{cr}$ when the tangent stiffness matrix of the geometric nonlinear equation of the structure becomes singular. This can be achieved by solving the following eigenvalue problem:(10)$$([K_{0}]+[K_{\sigma }]+[K_{\varepsilon }])\{\phi \}=0$$
where [*K*_0_] represents the elastic‒plastic stiffness matrix of the element; [*K**_σ_*] represents the initial stress matrix of the element; [*K**_ε_*] represents the initial strain matrix of the element;

The arc length method is an iterative method based on the modified Newton-Raphson method, which can comprehensively consider geometric nonlinearity and material nonlinearity. Considering geometric nonlinearity and material nonlinearity, the stiffness matrix is a function of the displacement of the structure. During each incremental step load, the structure stiffness was assumed to be constant. Through the multistage linear fitting method, the load‒displacement curve was fitted. When the load is close to the unstable ultimate load, the principal element of the total stiffness matrix of the structure is close to zero and stiffness matrix singularity occurs.

The detailed solution process is as follows: supposing the calculation result of the $n^{th}$ load step is on the real equilibrium path and the external load of the load step is known as $F_{n}$, the displacement is $u_{n}$, the reference load is $F_{0}$, and the load increment factor is $\Delta \lambda_{n}$. Then, we can find the load increment factor $\Delta \lambda_{n+1}$ of the ${\left(n+1\right)}^{th}$ load step with the given arc-length Δl.

Setting $u_{n+1}^{\left(0\right)}=u_{n}$ and using the modified Newton‒Raphson method, the iterative equation is as follows:(11)$$K_{n+1}^{\left(i\right)}\Delta u_{n+1}^{\left(i\right)}=\Delta {\tilde{F}}_{n+1}^{\left(i\right)}\;\Delta \lambda_{n+1}^{\left(i\right)}F_{0},$$
where $K_{n+1}^{\left(i\right)}=K_{n+1}^{\left(0\right)}=K_{n+1}^{\left(0\right)}\left(u_{n+1}^{\left(0\right)}\right)$, $\Delta {\tilde{F}}_{n+1}^{\left(i\right)}$ is referred to as the residual force of the unbalanced force in the $i^{th}$ iteration of the ${\left(n+1\right)}^{th}$ load step and is calculated as follows:(12)$$\Delta {\tilde{F}}_{n+1}^{\left(i\right)}=F_{n}+\lambda_{n+1}^{\left(i\right)}F_{0}-K\left(u_{k+1}^{\left(i\right)}\right)u_{k+1}^{\left(i\right)}$$
where $\lambda_{n+1}^{\left(i\right)}$ is the reference load factor of the $i^{th}$ iteration of the ${\left(n+1\right)}^{th}$ load step; $\Delta \lambda_{n+1}^{\left(i\right)}$ is the load increment factor, that is, the unknown quantity to be solved, and
(13)$$\lambda_{n+1}^{\left(i+1\right)}=\lambda_{n+1}^{\left(i\right)}-\Delta \lambda_{n+1}^{\left(i\right)}$$
(14)$$\lambda_{n+1}^{\left(0\right)}=\Delta \lambda_{n}$$
(15)$$\Delta \lambda_{n+1}^{\left(0\right)}=0$$
making
(16)$$\Delta u_{n+1}^{\left(i\right)}=\Delta u_{n+1,Ι}^{\left(i\right)}-\Delta \lambda_{n+1}^{\left(i\right)}\Delta u_{n+1,ΙΙ}^{\left(i\right)}$$
Substituting Equation (16) into Equation (11):(17)$$K_{n+1}^{\left(i\right)}\Delta u_{n+1,Ι}^{\left(i\right)}=\Delta {\tilde{F}}_{n+1}^{\left(i\right)}$$
(18)$$K_{n+1}^{\left(i\right)}\Delta u_{n+1,ΙΙ}^{\left(i\right)}=F_{0}$$

According to Equations (17) and (18), $\Delta u_{n+1,Ι}^{\left(i\right)}$ and $\Delta u_{n+1,ΙΙ}^{\left(i\right)}$ can be calculated. Then, substituting Equation (11) into the following equation we obtain
(19)$${\left(\Delta u_{n+1}^{\left(i\right)}\right)}^{T}\Delta u_{n+1}^{\left(i\right)}+{\left(\Delta \lambda_{n+1}^{\left(i\right)}F_{0}\right)}^{2}={\left(\Delta l\right)}^{2}$$

After simplification, the above equation can be written as follows:(20)$$\alpha_{1}{\left(\Delta \lambda_{n+1}^{\left(i\right)}\right)}^{2}-\alpha_{2}\Delta \lambda_{n+1}^{\left(i\right)}+\alpha_{3}=0$$
where $\alpha_{1}={\left(\Delta u_{n+1,Ι}^{\left(i\right)}\right)}^{T}\Delta u_{n+1,ΙΙ}^{\left(i\right)}$, $\alpha_{2}=2{\left(\Delta u_{n+1,Ι}^{\left(i\right)}\right)}^{T}\Delta u_{n+1,ΙΙ}^{\left(i\right)}$, and $\alpha_{3}={\left(\Delta u_{n+1,Ι}^{\left(i\right)}\right)}^{T}\Delta u_{n+1,Ι}^{\left(i\right)}-{\left(\Delta l\right)}^{2}$.

From the quadratic Equation (19), two roots of $\Delta \lambda_{n+1}^{\left(i\right)}$ can be obtained, and the positive roots will cause the iterative path to progress. If both roots are positive, the root closest to the linear solution $\alpha_{3}/\alpha_{2}$ is taken as $\Delta \lambda_{n+1}^{\left(i\right)}$. When conducting an analysis using the finite element method, the load step length and arc-length should be adjusted according to the different conditions of the members, so as to track the descending segment of the loading curve and find the limit point of the load‒displacement curve.

The width-thickness ratios of the sections are 10, 15, 20, 25, 30, 35, 40, and 50, respectively. The values of residual stress corresponding to different width-thickness ratios are as follows: when b/t ≤ 20, the tensile stress coefficient and compressive stress coefficient are 0.86, 0.22 and −0.175, −0.12 respectively; when b/t ≤ 30, the tensile stress coefficient and compressive stress coefficient are 0.8, 0.15 and −0.15, −0.1 respectively; when b/t ≤ 50, the tensile stress coefficient and compressive stress coefficient are 0.75, 0.145 and −0.135, −0.1, respectively. By imposing the residual stresses to the element integral point of the finite element model through a preprogrammed residual stress file, the cross-section axial stress cloud diagram can be obtained, as shown in Figure 13.

### 3.2. Numerical Simulation Results and Analysis

#### 3.2.1. Influence of Residual Stress on Stability with Different Width-Thickness Ratios 

Based on the finite element model that was established in the previous section, the stability factors of the axial compression member of a butt-welding box section with or without welding residual stress are calculated and shown in Figure 14. The black curve represents the influence of residual stress on the stability coefficient of axial compression members, while the red curve represents the influence of residual stress on the stability coefficient of axial compression members. The detailed results are shown in Figure 14a–j.

Figure 14 shows that the residual stress has a significant influence on the stability factor of the box section specimen with different width-thickness ratios. The general trend is that the influence of residual stress gradually decreases with an increase in the width-thickness ratio. The reason is that, with the increase in the width-thickness ratio, the buckling of members will predominantly be local elastic buckling. At the same time, it can be seen that the residual stress has different effects on the stability of members with different width-thickness ratios.

#### 3.2.2. Influence of Residual Stress on Stability Factor with Different Slenderness Ratios

As shown in Figure 14a, when the member slenderness is small, the influence of residual stress on the member’s stability does not exceed 5%. A small slenderness ratio reduces the influence of residual stresses on the stability, as shown in Figure 15. Figure 15a,c,e present the stress cloud diagrams without residual stress when b/t = 10, b/t = 30, and b/t = 50; Figure 15b,d,f are the equivalent stress cloud diagrams with residual stress when b/t = 10, b/t = 30, and b/t = 50.

Within a slenderness ratio range of 20–70, when b/t ≤ 35, the influence of the presence of residual stress on the member stability factor difference exceeds 5%; when b/t > 35, the difference is within 5%. The main reason is that when b/t ≤ 35, members with no residual stress exhibit local and global buckling modes, while members with residual stress exhibit global buckling modes, and members with or without residual stress exhibit different buckling modes. When b/t > 35, members present a local buckling mode whether there is residual stress or not, indicating that the residual stress effects are relatively small; details are shown in Figure 16 and Figure 17. Figure 16a,c,e,g,i,k,m and Figure 17a,c,e,g,i,k,m present the stress cloud diagrams without residual stress when b/t = 10, 15, 20, 25, 35, 40, and 50; Figure 16b,d,f,h,j,l,n and Figure 17b,d,f,h,j,l,n are the equivalent stress slenderness ratios of residual stress when b/t = 10, 15, 20, 25, 35, 40, and 50.

When slenderness ratios are larger than 70, whether there is residual stress or not, the difference is within 5%; all the members present the global buckling mode, as shown in Figure 18. It can also be seen in Figure 18 that when the slenderness ratio is more than 80 and the width-thickness ratio is 35, the member exhibits global and local correlation buckling. Then, with the increase in the width-thickness ratio, the post-buckling strength effect appears, resulting in the stability factor increasing instead of decreasing when the width-thickness ratio is more than 35. Figure 18a,c,e,g,i,k,m are axial stress cloud diagrams without residual stress when b/t = 10, 15, 20, 25, 35, 40, or 50; Figure 18b,d,f,h,j,l,n are the axial stress cloud diagrams of residual stress when b/t = 10, 15, 20, 25, 35, 40, or 50.

#### 3.2.3. Influence of Residual Stress on Stability Factor with Different Slenderness Ratios and Width-Thickness Ratios

In order to further investigate the influence of residual stress on the stability of members with different slenderness ratios and width-thickness ratios, the finite element calculation results were analyzed and the axial-displacement curves were obtained as shown in Figure 19. The black curves in Figure 19 are the axial-displacement curves without the influence of residual stress, while the red curves demonstrate the effect of residual stress.

As shown in Figure 19, when the flange width-thickness ratios are relatively small, the slope of the axial‒displacement curve is steeper, which indicates that the member has a large compression stiffness, and the member presents a global buckling instability mode. However, the residual stress has a great influence on the stability of the member, and the curve difference is evident with or without residual stress, especially when the slenderness ratio is small.

With the increase in the width-hickness ratio, the influence of residual stress on the stability of the member decreases, and the load axial displacement curves tend to be consistent with or without residual stress. This is due to the local buckling of the flange or web before the specimen reaches its ultimate bearing capacity. At the same time, it is also found that the axial‒displacement curve presents a more obvious descending period after passing the extreme value, which reduces the compression stiffness of the short column.

The black curves in Figure 20 do not consider the influence of residual stress, while the red curves consider the influence of residual stress.

The slenderness ratio producing the biggest difference is usually the optimal slenderness ratio according to the equal stability, and the detailed analysis process is as follows.

The equal stability design is where the local stability of the plate is consistent with the global elastic stability of the member. The local stress of the thin-walled box section plate of high-strength steel is given as follows:(21)$$\sigma_{cr}=\frac{k\pi^{2}E}{12(1-\upsilon^{2})}{\left(\frac{t}{b}\right)}^{2}$$
where *k* is the clamping coefficient of both ends of the plate. The global stability coefficient of the members is as follows:(22)$$\sigma =\frac{\pi^{2}E}{\lambda^{2}}$$
when $\sigma =\sigma_{cr}$, the following equation can be obtained:(23)$$\lambda =\sqrt{\frac{12(1-v^{2})}{k}}(b/t)$$

It is difficult to determine the constraints’ clamping coefficient. Due to the presence of initial defects, one side of the web will be in tension and the other side will be in compression. For practical cases, at the compression side, the web is constrained by the upper and lower flange. Its clamping coefficient is between one fixed end and one free end (*k* = 1.28) and simply supported on both ends (*k* = 4) [38]. Taking the median value of 2.64 and substituting it into Equation (23), the slenderness ratio under different width-thickness ratios can be obtained, as shown in Table 7.

It can be seen from Table 7 and Figure 20g that the stability of members is relatively sensitive to residual stress in the equal stability design. When the width-thickness ratio is larger than 30, the difference of the stability factor corresponding to each slenderness ratio is within 5%, and the residual stress has little influence on the stability factor of the members.

## 4. Conclusions

In this paper, the residual stress of a BS700 butt-welded box section axial compression member is studied using the blind-hole method. A finite element model considering initial geometric defects and residual stress was established, and the influence of residual stress on the stability of axially compressed members was analyzed. The following conclusions were obtained:(1)The longitudinal tensile stress is the main welding residual stress of butt-welding box-shaped section members, with an average peak residual tensile stress of 570.1 MPa and a maximum peak stress of 619.8 MPa, which has a great impact on the stability of the structure. In practical engineering applications, the residual stress in welded structures should be eliminated.(2)The influence of residual stress on the stability coefficient of axial compression members decreases with an increase in the width-thickness ratio. In the design of a box section, the width-thickness ratio should be appropriately increased to reduce the influence of residual stress.(3)When the slenderness ratio of members is between 20 and 70, the residual stress has a greater influence on the stability coefficient of the members under axial compression, while the influence outside the interval is relatively small.(4)According to the analysis, in the case of equal stability design, the residual stress has a great influence on the stability of the member. Therefore, the influence of residual stress should be considered in the stability design of the member.

## Figures and Tables

**Figure 1 materials-13-03282-f001:**
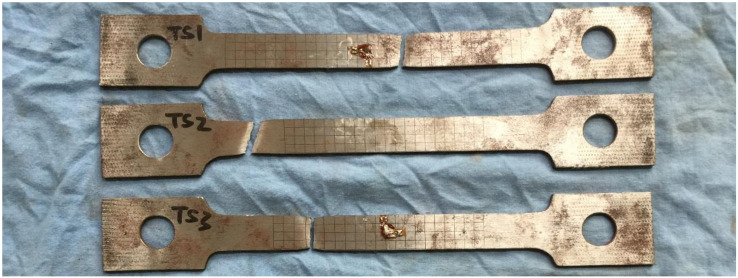
Specimens of tensile testing after failure.

**Figure 2 materials-13-03282-f002:**
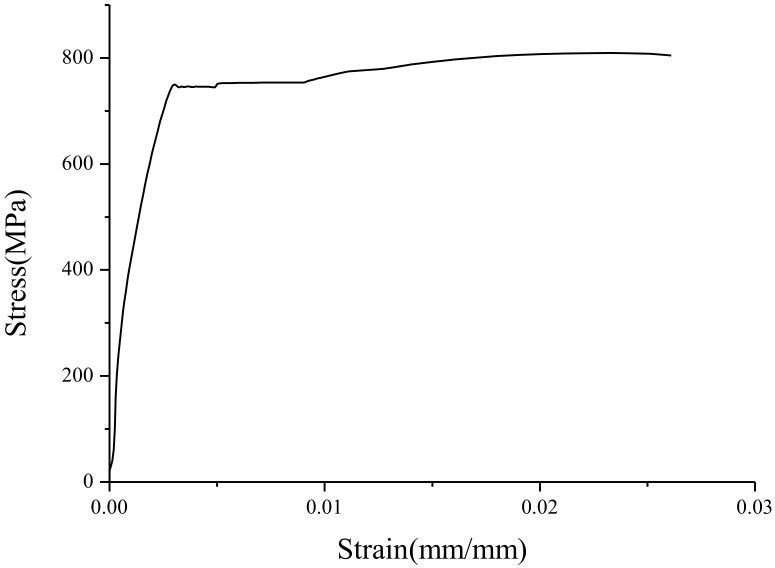
Stress-strain curve of the BS700 high-strength steel TS1 specimen.

**Figure 3 materials-13-03282-f003:**
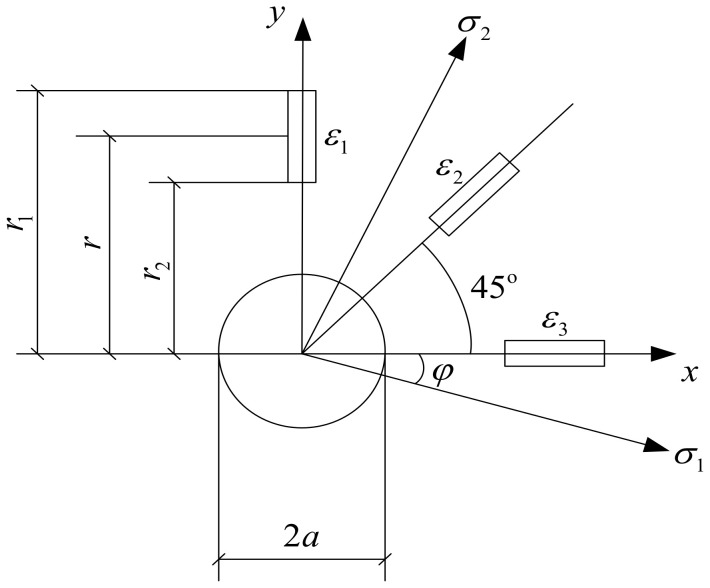
Principle of blind-hole drilling method.

**Figure 4 materials-13-03282-f004:**
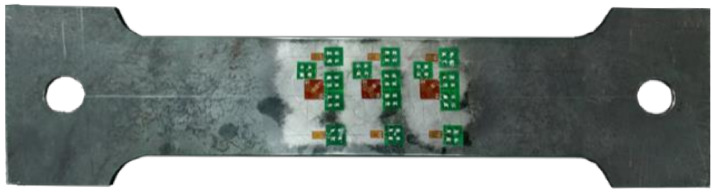
Calibration specimen strain rosette’s layout.

**Figure 5 materials-13-03282-f005:**
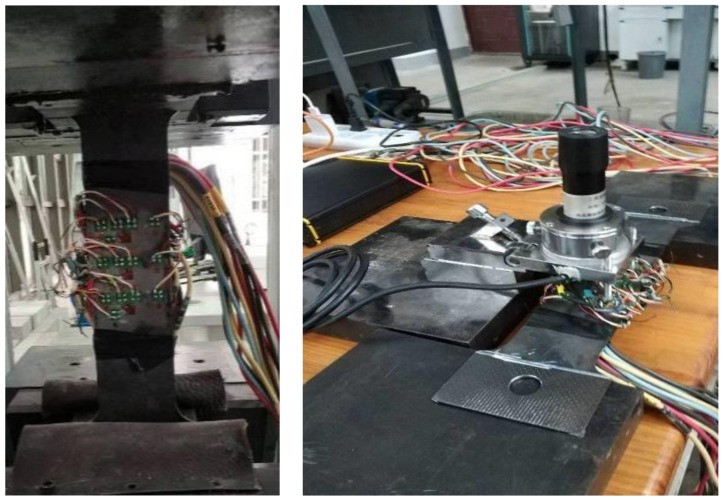
Calibration coefficient test.

**Figure 6 materials-13-03282-f006:**
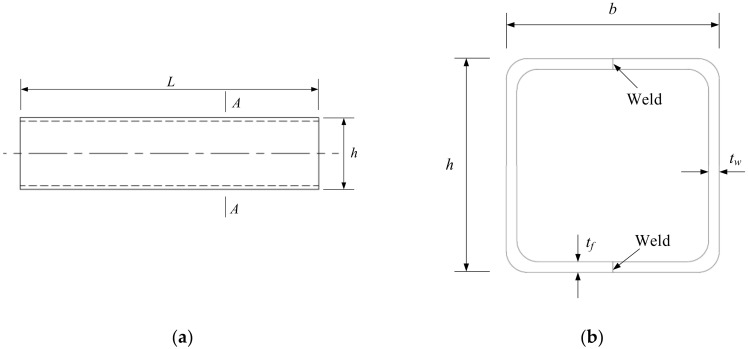
Drawings of samples. (**a**) Layout of the samples (**b**) Section A-A.

**Figure 7 materials-13-03282-f007:**
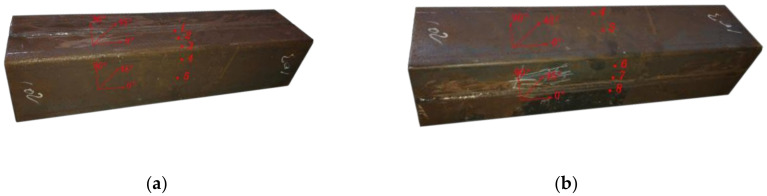
Test point location diagram of the welded box section RS1 specimen. (**a**) Points 1 to 4, (**b**) Points 5 to 8.

**Figure 8 materials-13-03282-f008:**
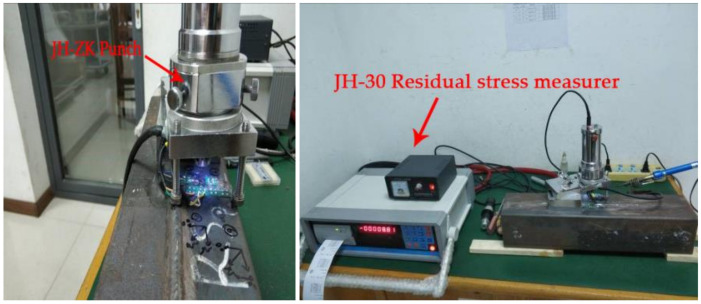
Residual stress measurement test.

**Figure 9 materials-13-03282-f009:**
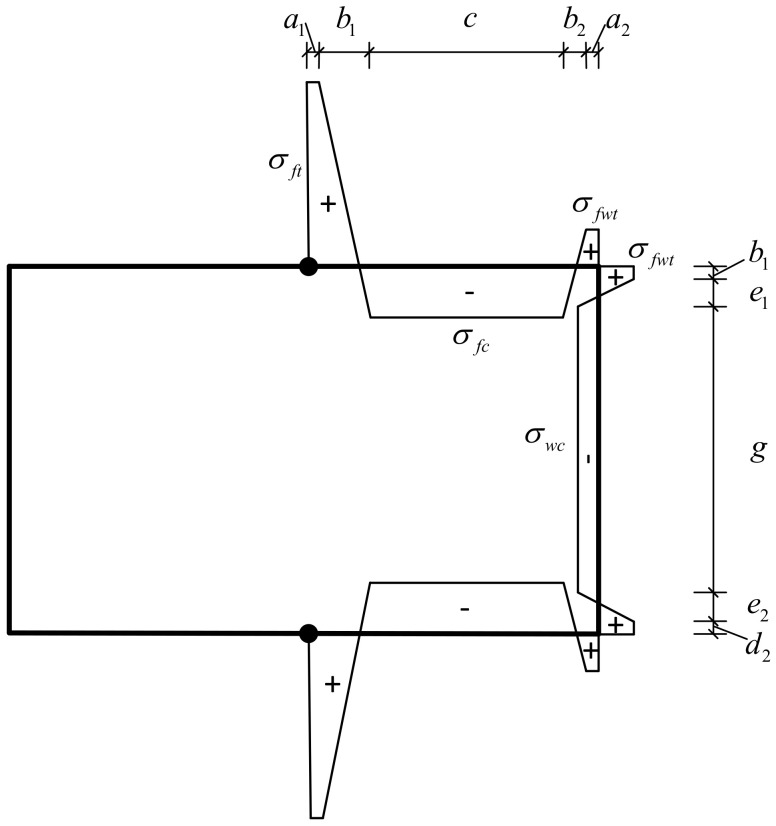
Residual stress model.

**Figure 10 materials-13-03282-f010:**
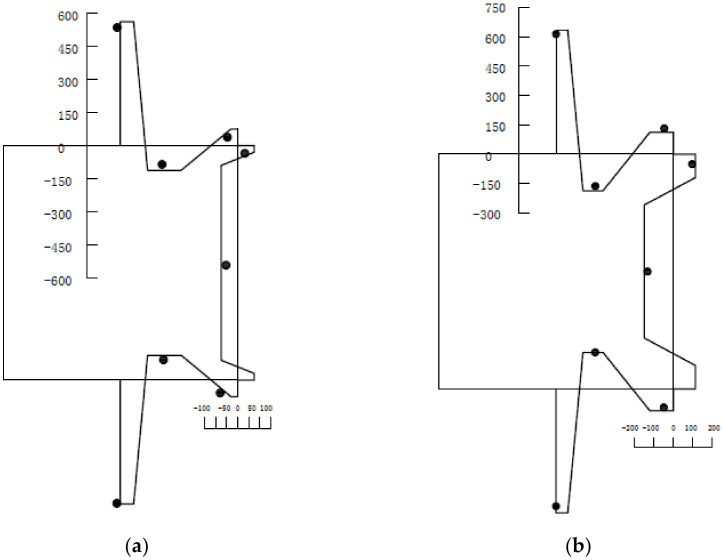
Residual stress model and measured data (Unit: MPa): (**a**) RS1, (**b**) RS4.

**Figure 11 materials-13-03282-f011:**
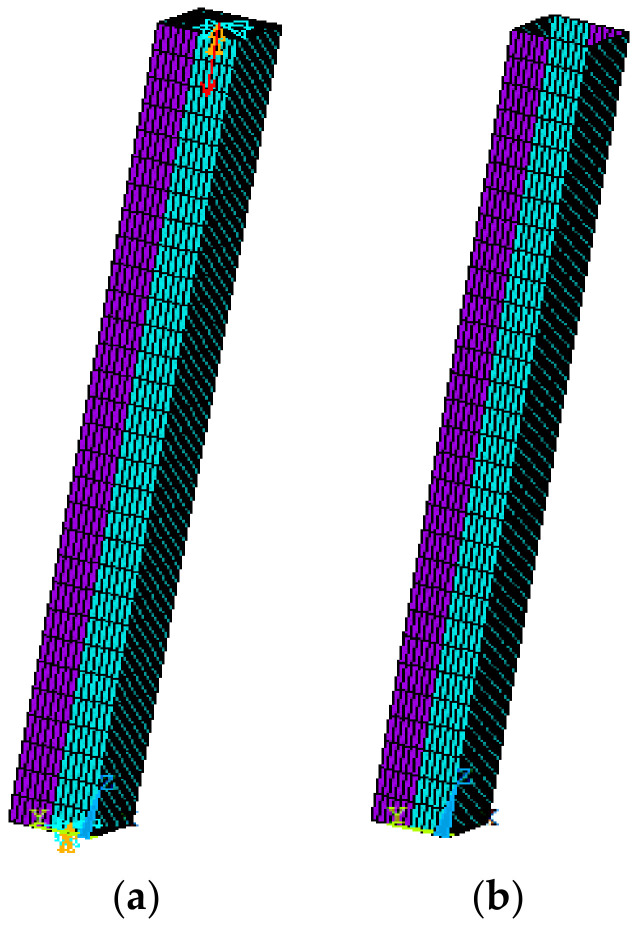
Basic finite element model. (**a**)with constraint, (**b**) without constraint

**Figure 12 materials-13-03282-f012:**
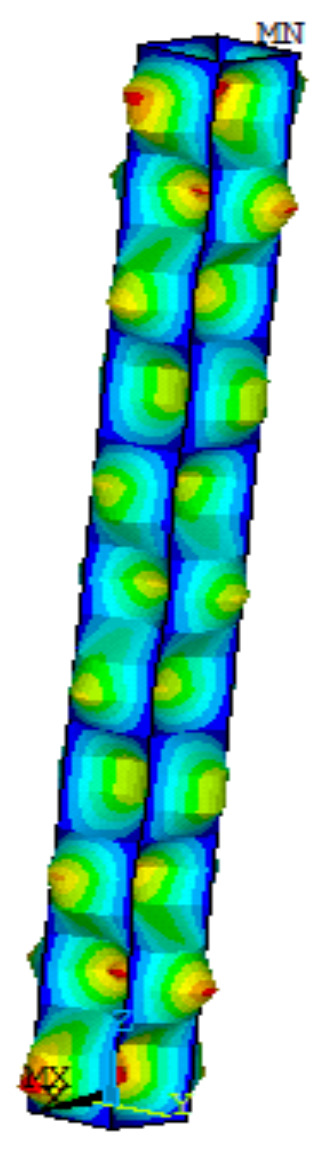
Local buckling modes of members.

**Figure 13 materials-13-03282-f013:**
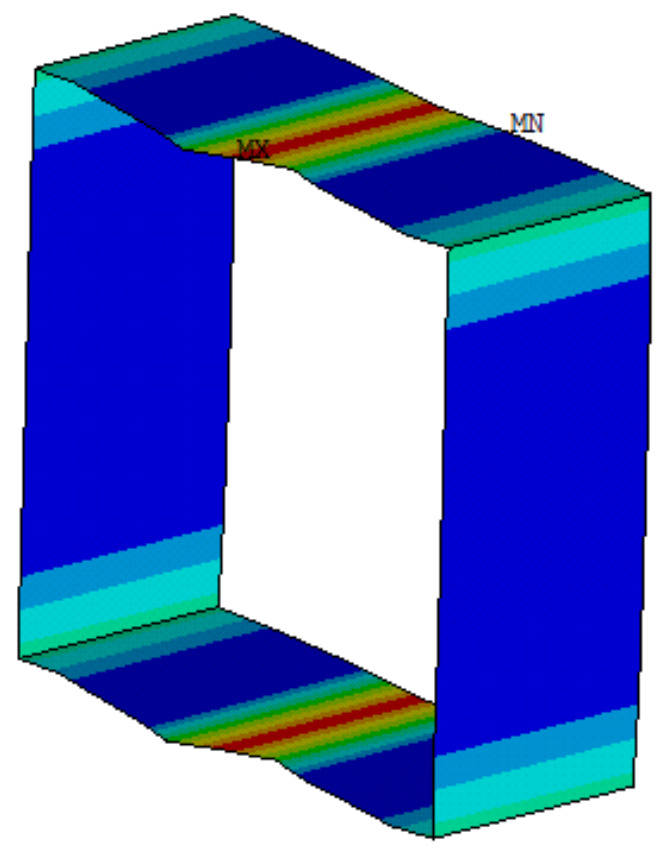
Stress cloud diagram of a section after applying residual stress.

**Figure 14 materials-13-03282-f014:**
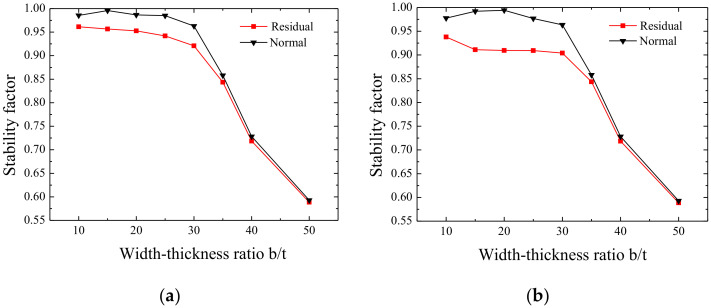
Influence of different width‒thickness ratios on residual stress: (**a**) λ = 15, (**b**) λ = 20, (**c**) λ = 30, (**d**) λ = 40, (**e**) λ = 50, (**f**) λ = 60, (**g**) λ = 70, (**h**) λ = 80, (**i**) λ = 90, and (**j**) λ = 100.

**Figure 15 materials-13-03282-f015:**
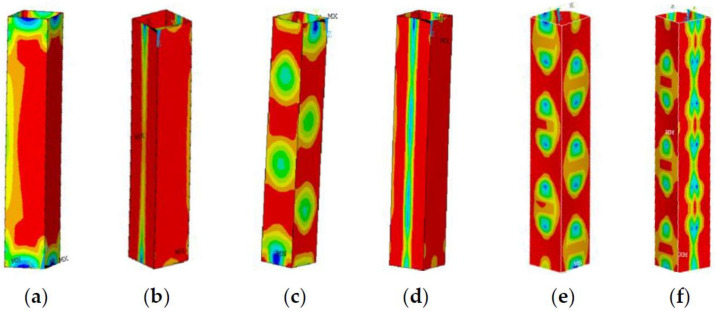
The buckling equivalent stress cloud diagram when λ = 15. (**a**) b/t = 10, (**c**), b/t = 30, and (**e**) b/t = 50 without residual stress; (**b**) b/t = 10, (**d**) b/t = 30, and (**f**) b/t = 50 with residual stress.

**Figure 16 materials-13-03282-f016:**
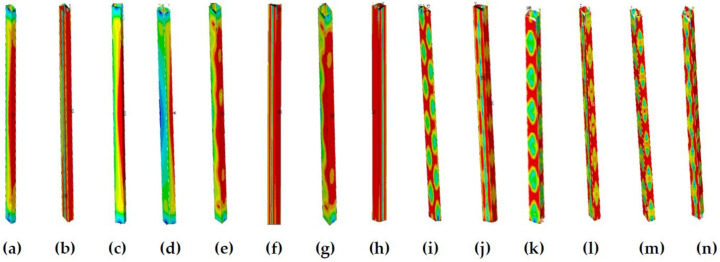
The buckling equivalent stress cloud diagram when λ = 30. (**a**) b/t = 10, (**c**), b/t = 15, (**e**) b/t = 20, (**g**) b/t = 25, (**i**) b/t = 35, (**k**) b/t = 40, and (**m**) b/t = 50 without residual stress; (**b**) b/t = 10, (**d**) b/t = 15, (**f**) b/t = 20, (**h**) b/t = 25, (**j**) b/t = 35, (**l**) b/t = 40, and (**n**) b/t = 50 with residual stress.

**Figure 17 materials-13-03282-f017:**
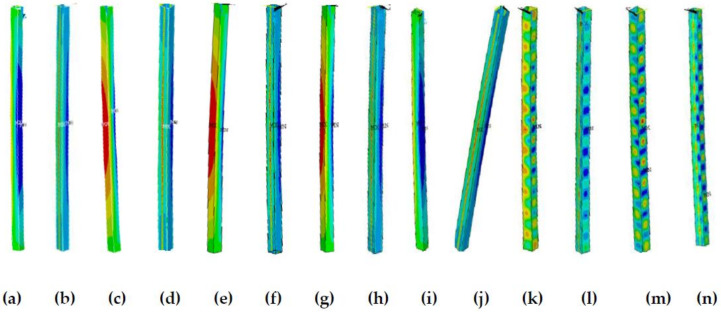
The buckling equivalent stress cloud diagram when λ = 60. (**a**) b/t = 10, (**c**), b/I5, (**e**) b/t = 20, (**g**) b/t = 25, (**i**) b/t = 35, (**k**) b/t = 40, and (**m**) b/t = 50 without residual stress; (**b**) b/t = 10, (**d**) b/t = 15, (**f**) b/t = 20, (**h**) b/t = 25, (**j**) b/t = 35, (**l**) b/t = 40, and (**n**) b/t = 50 with residual stress.

**Figure 18 materials-13-03282-f018:**
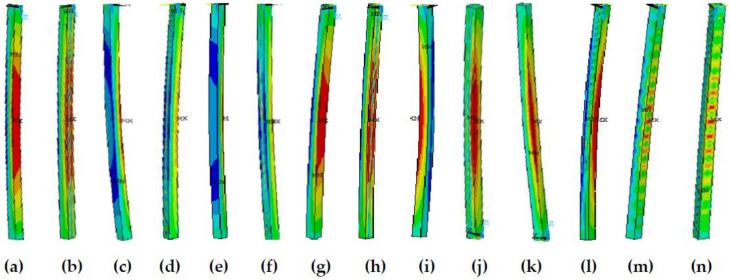
The buckling equivalent stress cloud diagram when λ = 80. (**a**) b/t = 10, (**c**), b/t = 15, (**e**) b/t = 20, (**g**) b/t = 25, (**i**) b/t = 35, (**k**) b/t = 40, and (**m**) b/t = 50 without residual stress; (**b**) b/t = 10, (**d**) b/t = 15, (**f**) b/t = 20, (**h**) b/t = 25, (**j**) b/t = 35, (**l**) b/t = 40, and (**n**) b/t = 50 with residual stress.

**Figure 19 materials-13-03282-f019:**
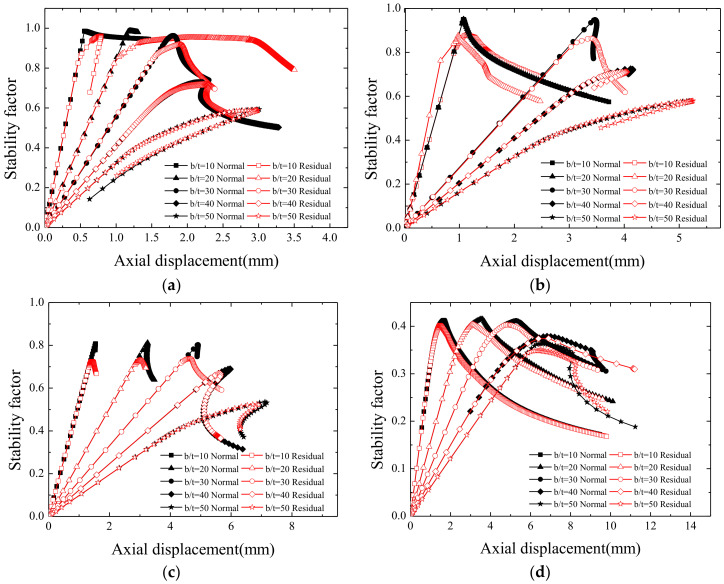
Load axial displacement curves with different slenderness and width-thickness ratios. (**a**) λ = 15, (**b**) λ = 30, (**c**) λ = 50, and (**d**) λ = 80.

**Figure 20 materials-13-03282-f020:**
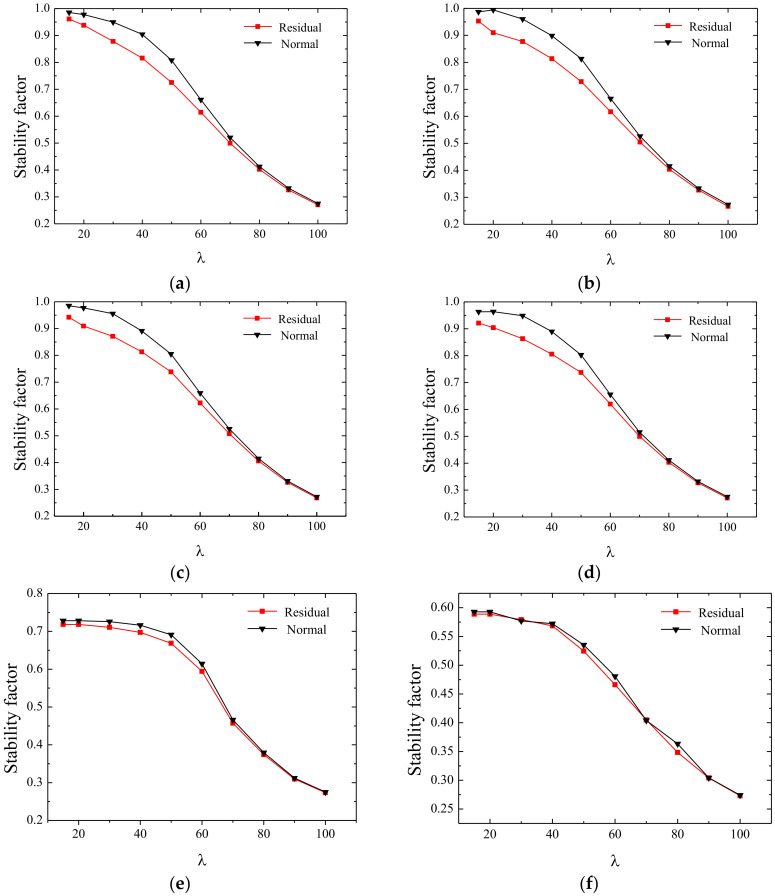
Effect of residual stress on the stability factor when the width-thickness ratio is certain. (**a**) b/t = 10, (**b**) b/t = 20, (**c**) b/t = 25, (**d**) b/t = 30, (**e**) b/t = 40, (**f**) b/t = 50, and (**g**) percentage difference of stability factor with or without residual stress.

**Table 1 materials-13-03282-t001:** Chemical composition of experimental BS700 hot-rolled plates.

Material	C	Si	Mn	P	S	Ti	Nb	V	Fe
BS700MC	≤0.10	≤0.40	≤0.10	0.014	≤0.10	≤0.10	≤0.10	≤0.10	Base

**Table 2 materials-13-03282-t002:** Geometric parameters of specimens.

Specimens	L (mm)	B (mm)	t (mm)
TS1	200.11	20.04	3.18
TS2	200.04	20.15	3.15
TS3	200.16	20.04	3.10

**Table 3 materials-13-03282-t003:** Basic mechanical properties of BS700 high-strength steel.

Specimens	Young’s Modulus E (GPa)	Yield Stress f_y_ (MPa)	Ultimate Stress f_u_ (MPa)
TS1	208.7	746.7	807.3
TS3	207.1	742.1	778.9
Average	207.9	744.4	793.1

**Table 4 materials-13-03282-t004:** Butt size parameters of the cross-section design.

Specimen Number	Flange Width b/mm	Flange Thickness t_f_/mm	b/t_f_	Web Height h/mm	Web Thickness t_w_ /mm	h/t_w_	Length L/mm
RS1	105.9	3.68	28.8	106.1	3.25	32.6	450
RS2	60.3	3.21	18.8	60	3.76	16.0	250
RS3	210.2	6.46	32.5	209.8	6.23	33.7	660
RS4	120.4	6.15	19.6	120.1	5.96	20.2	400

**Table 5 materials-13-03282-t005:** Residual stress values of each test point (MPa).

Specimen Number	Point 1 σ_ft1_	Point 2 σ_r2_	Point 3 σ_fwt1_	Point 4 σ_fwt2_	Point 5 σ_wc_	Point 6 σ_fwt3_	Point 7 σ_r3_	Point 8 σ_ft2_
RS1	542.1	−98.2	46.3	32.6	−62.3	65.1	−82.7	564.4
RS2	589.2	52.1	119.4	108.7	−152.8	98.5	25.4	601.5
RS3	513.2	−98.4	81.5	71.6	−81.2	68.3	−79.8	538.4
RS4	619.8	−162.1	128.2	103.9	−141.3	76.4	−185.1	592.3

**Table 6 materials-13-03282-t006:** Residual stress distribution pattern data of specimens.

Specimen Number	σ_ft_/MPa	σ_fwt_/MPa	σ_fc_/MPa	σ_wc_/MPa	a_1_/mm	b_1_/mm	a_2_/mm	d_1_/mm
RS1	560.0	74.7	−112.0	−74.7	6	5.3	3	3
RS4	634.7	112.0	−186.7	−149.3	6	3	12	12

**Table 7 materials-13-03282-t007:** Slenderness ratio of equal stability design.

Width–Thickness Ratio (b/t)	Slenderness Ratio (λ)
10	20.3
15	30.5
20	40.7
25	50.8
30	61.0

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
