# Peer review of "Research on Residual Stress of a BS700 Butt-Welded Box Section and Its Influence on the Stability of Axial Compression Members"

_materials, 2020, doi:10.3390/ma13153282_

Round 1
Reviewer 1 Report
The article is very interesting. The problem of residual stresses in welded sections was discussed. Compression tests of 4 welded box samples were carried out. Calculation were performed in the program based on the finite element method. Practical conclusions were made. However, the reviewer has a few comments:
- Fig. 2 shows the stress-strain diagram. Is it a test graph of any of the samples TS1, TS2, TS3 or an average graph? This should be explained in the text.
- Destruction of the TS2 sample occurred outside the measuring range due to improper installation in the jaws of the testing machine. According to the reviewer, this result should be excluded. In Table 3, the average z should be calculated only from the results of testing the TS1 and TS3 samples.
- Please provide drawings of samples RS1-RS4 with dimensions.
- What program was used for FEM analysis? Please complete this information.
- What material model was adopted? Huber-Miles-Henkcy yield Surface? Did you consider the plasticity of the material? Please complete this information.
- An additional reference to the following publications is suggested:
- Li, C.; Dong, S.; Wang, T.; Xu, W.; Zhou, X. Numerical Investigation on Ultimate Compressive Strength of Welded Stiffened Plates Built by Steel Grades of S235–S390. Sci. 2019, 9, 2088. https://doi.org/10.3390/app9102088
- Chiocca, A.; Frendo, F.; Bertini, L. Evaluation of Heat Sources for the Simulation of the Temperature Distribution in Gas Metal Arc Welded Joints. Metals 2019, 9, 1142. https://doi.org/10.3390/met9111142
Author Response
Journal of Materials
Manuscript Number: 852361
Title: Research on Residual Stress of a BS700 Butt-Welded Box Section and Its Influence on the Stability of Axial Compression Members
We would like to thank the reviewers for reviewing our manuscript and giving valuable comments and suggestions. Based on these comments, the manuscript has been revised accordingly. For editor and reviewers’ easy reference, the reviewers’ comments, followed with our answers and revisions are listed below.
Reviewer #1:
The article is very interesting. The problem of residual stresses in welded sections was discussed. Compression tests of 4 welded box samples were carried out. Calculation were performed in the program based on the finite element method. Practical conclusions were made. However, the reviewer has a few comments:
- Fig. 2 shows the stress-strain diagram. Is it a test graph of any of the samples TS1, TS2, TS3 or an average graph? This should be explained in the text.
- Figure 2 shows the stress-strain curve of TS1 specimen. It has been added in the manuscript(In line 101)
- Destruction of the TS2 sample occurred outside the measuring range due to improper installation in the jaws of the testing machine. According to the reviewer, this result should be excluded. In Table 3, the average z should be calculated only from the results of testing the TS1 and TS3 samples..
- The result of TS2 has been excluded, the average z calculate from the results of testing TS1 and TS3 samples. The result has been provided in Table3. Therefore, we have updated the percentage of residual stress in the abstract and section 2.3. The update content is shown in line 19, 21, 182 and 184.
- Please provide drawings of samples RS1-RS4 with dimensions.
- Drawings of samples have been added as Figure 6.
- What program was used for FEM analysis? Please complete this information.
- FEM analysis information has been added in the manuscript(From line 213 to 214).
5.What material model was adopted? Huber-Miles-Henkcy yield Surface? Did you consider the plasticity of the material? Please complete this information.
- The relevant information has been added in the manuscript(From line 216 to 218).
- An additional reference to the following publications is suggested.
- Reference articles have been added in the manuscript(From line 67 to 70).

Reviewer 2 Report
The theme of the papers is very interesting and useful for the industrial application.
Description of sample preparation missing. What kind of welding were the seams made of?
110: On the Fig.2. the "strain" unit is not indicated. It is not clear which sample is presented on the figure.
156: Table 4. include the measurements point of which sample??
The structure of the article is difficult to interpret because the results are not clearly articulated.
If the theoretical calculations were be supported by the experiment would be even more valuable the article.
After correction the paper can be acceptable.
Author Response
Journal of Materials
Manuscript Number: 852361
Title: Research on Residual Stress of a BS700 Butt-Welded Box Section and Its Influence on the Stability of Axial Compression Members
We would like to thank the reviewers for reviewing our manuscript and giving valuable comments and suggestions. Based on these comments, the manuscript has been revised accordingly. For editor and reviewers’ easy reference, the reviewers’ comments, followed with our answers and revisions are listed below.
Reviewer #2:
The theme of the papers is very interesting and useful for the industrial application:
- Description of sample preparation missing. What kind of welding were the seams made of.
- The relevant information has been added in the manuscript(From line 97 to 98).
- 110: On the Fig.2. the "strain" unit is not indicated. It is not clear which sample is presented on the figure.
- Figure 2 shows the stress-strain curve of TS1 specimen. The relevant information has been added in the manuscript(In line 111).
- 156: Table 4. include the measurements point of which sample?? The structure of the article is difficult to interpret because the results are not clearly articulated.
- Table 4 include the measurements point of samplesRS1-RS4. Drawings of samples have been added in Figure 6 and the position of measuring point have been shown in Figure 7.
- This paper is organized as follows: details on the material properties and the residual stress test are presented in Section 2, along with a residual stress distribution model of a butt-welding box section. In Section 3, through a numerical simulation, the residual stress distribution law of welding members and its effect on the stability of the axial compression members are analyzed. In Section 4, some conclusions and suggestions for engineering are given.
- If the theoretical calculations were be supported by the experiment would be even more valuable the article. After correction the paper can be acceptable.
- In this study, the residual stress of a BS700 butt-welded box section axial compression members was studied by the blind-hole method, its distribution law was summarized, and a residual stress distribution model was established. By establishing a finite-element model considering the initial geometric imperfection and residual stress, the influence of residual stress on the stability of axial compression members was analyzed. Although we have done the residual stress related experiments, we also believe that the in-depth experimental verification of the stability numerical calculation model is very important for this research direction, this content will be our next research in the future.

Reviewer 3 Report
The article experimentally and numerically investigates the influence of the welding residual stress on the stability of compressed members.
The Authors clearly describe the test set-up implementation.
The paper is clearly written and it contributes to our further knowledge on this topic.
It is indeed well in line with the journal aims and scope and deserves publication.
However, there are some minor issues that should be addressed prior to publication, listed below:
- Section 3: the authors should specify if the numerical analyses were performed using an hand-made finite element program or commercial software.
- Lines 212: "shell181" at which finite element type refers?
- Fig. 11-12-15-16-17: in these figures a contour plot is showed, but the colour-value legend should be specified.
- In Fig. 18 the load-displacement curves should be represented, according to what is reported in the text. Neverthless, in the figure the stability factor-displacement is reported in the graphs. Probably the text and the figure should be conceptually aligned.
Author Response
Journal of Materials
Manuscript Number: 852361
Title: Research on Residual Stress of a BS700 Butt-Welded Box Section and Its Influence on the Stability of Axial Compression Members
We would like to thank the reviewers for reviewing our manuscript and giving valuable comments and suggestions. Based on these comments, the manuscript has been revised accordingly. For editor and reviewers’ easy reference, the reviewers’ comments, followed with our answers and revisions are listed below.
Reviewer #3:
The article experimentally and numerically investigates the influence of the welding residual stress on the stability of compressed members. The Authors clearly describe the test set-up implementation.
The paper is clearly written and it contributes to our further knowledge on this topic. It is indeed well in line with the journal aims and scope and deserves publication. However, there are some minor issues that should be addressed prior to publication, listed below:
- Section 3: the authors should specify if the numerical analyses were performed using an hand-made finite element program or commercial software.
- Commercial software named ANSYS was used to establish a finite element, relevant information has been added in the manuscript(From line 213 to 214).
- Lines 212: "shell181" at which finite element type refers?
- "shell181" is a finite element model in ANSYS software which was described from line 216 to 217.
- Fig. 11-12-15-16-17: in these figures a contour plot is showed, but the colour-value legend should be specified.
- In ANSYS software, the warm color area indicates high stress, while the cool color area indicates low stress. Because this manuscript involves many such figures, it would be redundant to elaborate on each one.
- In Fig. 18 the load-displacement curves should be represented, according to what is reported in the text. Neverthless, in the figure the stability factor-displacement is reported in the graphs. Probably the text and the figure should be conceptually aligned.
- There is indeed a discrepancy between the text and the Figures description, and we have corrected the content in lines 354,355,362 and 369.

Reviewer 4 Report
The article is interesting, and provides new information, being of interest for being published. However, for its acceptance it is necessary to make a major revision.
- In line 157 de points to measure is described but could be useful to have in a little scheme where they could be seen easily.
- Figure 7. Indicate the different instruments in the picture
- In point 3, deeper information about the model is needed
Author Response
Journal of Materials
Manuscript Number: 852361
Title: Research on Residual Stress of a BS700 Butt-Welded Box Section and Its Influence on the Stability of Axial Compression Members
We would like to thank the reviewers for reviewing our manuscript and giving valuable comments and suggestions. Based on these comments, the manuscript has been revised accordingly. For editor and reviewers’ easy reference, the reviewers’ comments, followed with our answers and revisions are listed below.
Reviewer #4:
The article is interesting, and provides new information, being of interest for being published. However, for its acceptance it is necessary to make a major revision:
- In line 157 de points to measure is described but could be useful to have in a little scheme where they could be seen easily.
- In order to make the readers know the exact position of each measuring point clearly, the author suggests to keep the detailed description of the measuring point.
- Figure 7. Indicate the different instruments in the picture.
- The name of the instrument has been added in Figure 8(in line 170), These instruments are all manufactured in China.
- In point 3, deeper information about the model is needed.
- Information about the model has been added in the manuscript(From line 216 to 222).

Round 2
Reviewer 4 Report
The authors have correctly solved the questions asked.